# High-throughput Interpretation of Killer-cell Immunoglobulin-like Receptor Short-read Sequencing Data with PING

**Wesley M. Marin**[1], **Ravi Dandekar**[1], **Danillo G. Augusto**[1], **Tasneem Yusufali**[1], **Bianca Heyn**[2], **Jan Hofmann**[3], **Vinzenz Lange**[2], **Jürgen Sauter**[3], **Paul J. Norman**[4], **Jill A. Hollenbach**[1]*

**1** UCSF Weill Institute for Neurosciences, Department of Neurology, University of California, San Francisco, San Francisco, California, United States of America, **2** DKMS Life Science Lab, Dresden, Germany, **3** DKMS, Tübingen, Germany, **4** Division of Biomedical Informatics and Personalized Medicine, and Department of Immunology and Microbiology, University of Colorado Anschutz Medical Campus, Aurora, Colorado, United States of America

* jill.hollenbach@ucsf.edu

**Data Availability Statement:** Scripts used to process genotype data and generate figures are available from https://github.com/wesleymarin/

## Abstract

The *killer-cell immunoglobulin-like receptor* (*KIR*) complex on chromosome 19 encodes receptors that modulate the activity of natural killer cells, and variation in these genes has been linked to infectious and autoimmune disease, as well as having bearing on pregnancy and transplant outcomes. The medical relevance and high variability of *KIR* genes makes short-read sequencing an attractive technology for interrogating the region, providing a high-throughput, high-fidelity sequencing method that is cost-effective. However, because this gene complex is characterized by extensive nucleotide polymorphism, structural variation including gene fusions and deletions, and a high level of homology between genes, its interrogation at high resolution has been thwarted by bioinformatic challenges, with most studies limited to examining presence or absence of specific genes. Here, we present the PING (Pushing Immunogenetics to the Next Generation) pipeline, which incorporates empirical data, novel alignment strategies and a custom alignment processing workflow to enable high-throughput *KIR* sequence analysis from short-read data. PING provides *KIR* gene copy number classification functionality for all *KIR* genes through use of a comprehensive alignment reference. The gene copy number determined per individual enables an innovative genotype determination workflow using genotype-matched references. Together, these methods address the challenges imposed by the structural complexity and overall homology of the *KIR* complex. To determine copy number and genotype determination accuracy, we applied PING to European and African validation cohorts and a synthetic dataset. PING demonstrated exceptional copy number determination performance across all datasets and robust genotype determination performance. Finally, an investigation into discordant genotypes for the synthetic dataset provides insight into misaligned reads, advancing our understanding in interpretation of short-read sequencing data in complex genomic regions. PING promises to support a new era of studies of KIR polymorphism, delivering high-resolution *KIR* genotypes that are highly accurate, enabling high-quality, high-throughput *KIR* genotyping for disease and population studies.

ping_paper_scripts. PING pipeline code is available from https://github.com/wesleymarin/ping. All relevant validation genotype data are within the manuscript and its Supporting Information files. Synthetic sequence data is available from https://github.com/wesleymarin/KIR_synthetic_data.

**Funding:** This work was supported by the National Institutes of Health (https://www.nih.gov/) (NIH-R01AI128775). Award recipients are JAH, PJN and WMM. The funders had no role in study design, data collection and analysis, decision to publish, or preparation of the manuscript.

**Competing interests:** The authors have declared that no competing interests exist.

## Author summary

Killer cell immunoglobulin-like receptors (KIR) serve a critical role in regulating natural killer cell function. They are encoded by highly polymorphic genes within a complex genomic region that has proven difficult to interrogate owing to structural variation and extensive sequence homology. While methods for sequencing *KIR* genes have matured, there is a lack of bioinformatic support to accurately interpret *KIR* short-read sequencing data. The extensive structural variation of *KIR*, both the small-scale nucleotide insertions and deletions and the large-scale gene duplications and deletions, coupled with the extensive sequence similarity among *KIR* genes presents considerable challenges to bioinformatic analyses. PING addressed these issues through a highly-dynamic alignment workflow, which constructs individualized references that reflect the determined copy number and genotype makeup of a sample. This alignment workflow is enabled by a custom alignment processing pipeline, which scaffolds reads aligned to all reference sequences from the same gene into an overall gene alignment, enabling processing of these alignments as if a single reference sequence was used regardless of the number of sequences or of any insertions or deletions present in the component sequences. Together, these methods provide a novel and robust workflow for the accurate interpretation of *KIR* short-read sequencing data.

## Introduction

The *killer cell immunoglobulin-like receptor* (*KIR*) complex, located in human chromosomal region 19q13.42, encodes receptors expressed on the surface of natural killer (NK) cells [1] and a subtype of T-cells [2]. KIRs interact with their cognate HLA class I ligands to educate NK cells and modulate their cytotoxicity [3–5]. *KIR* genes exhibit presence and absence polymorphism and gene content variation that has been implicated in numerous immune-mediated and infectious diseases [6–11]. In addition, careful consideration of *KIR* gene content haplotypes for allogeneic transplantation has been shown to improve outcomes for acute myelogenous leukemia patients [12–17]. Whereas evidence for the relevance of *KIR* variation in health and disease is mounting, analysis of the *KIR* family at allelic resolution has been thwarted by the complexity of the region.

The *KIR* complex evolved rapidly through recombination and gene duplication events, and in humans this has resulted in a gene-content variable cluster of 13 genes and 2 pseudogenes [18–20]. Variation in *KIR* genes is characterized by extensive nucleotide polymorphisms, with 1110 alleles described to date [21]. The *KIR* complex is also characterized by large-scale structural variation, including gene fusions, duplications and deletions [22,23]. *KIR* haplotypes exhibit gene content variation at extraordinary levels, generating hundreds of observed haplotype structures [20,24–26].

The high variability of *KIR* makes short-read sequencing an attractive technology for interrogating the region, providing a high-throughput, high-fidelity and cost-effective sequencing method [27]. Whereas the *KIR* region is relatively small, between 70-270Kbp [28], the overall sequence similarity among genes, structural variability of the region, and sequence polymorphism present major obstacles to bioinformatics workflows. The high potential for read misalignments significantly confounds interpretation of the region in modern large-scale sequencing studies.

Previously, we introduced a laboratory method for targeted sequencing of the *KIR* gene complex [27], but the associated prototype bioinformatic pipeline for sequence interpretation

presented significant workload barriers for high-throughput studies. For example, the copy number determination workflow was unable to differentiate *KIR2DL2* from *KIR2DL3*, which are sets of highly similar allelic groups of the *KIR2DL23* gene [29], and the resolution of *KIR2DS1* and *KIR2DL1* was less precise than desired due to read misalignments caused by the similarity of these two genes. Additionally, a high frequency of unresolved genotypes (not matching any described allele sequence) necessitated subsequent interpretation by a user with domain expertise. In spite of these challenges, the prototype pipeline has provided insight into *KIR* genotyping methods development [30,31], the role of *KIR* sequence variants in immune dysfunction [17,32], and *KIR* evolutionary analyses [33]. To the best of our knowledge there are two other existing tools for interrogating *KIR* short-read sequencing data, KIR*IMP [34] and KPI [35,36]. KIR*IMP imputes *KIR* copy number from carefully selected SNPs while KPI interprets *KIR* gene content and predicts haplotype-pairs using *in silico* probes. Neither of these methods support allele level genotyping or direct copy number assessment.

Here, we present a comprehensive *KIR* sequence interpretation workflow, termed PING (Pushing Immunogenetics to the Next Generation), which builds on our early work by incorporating empirical optimizations derived from sequencing thousands of samples, in addition to novel alignment strategies to address issues with read misalignments, to provide a comprehensive *KIR* sequence analysis tool, offering allele-level genotypes, copy number, and novel sequence analysis. This work improves on the pipeline described in Norman et al. [27] in the following ways: the copy number determination workflow was adjusted from a single-sequence per gene alignment to a comprehensive multiple-sequence per gene alignment; virtual probes used for gene content determination were refined and expanded; the genotype determination workflow was adjusted from static single-gene filtration alignments to dynamic holistic alignments, which incorporate the so-established gene content and preliminary genotype determinations; a custom alignment processing workflow was developed to handle multiple-sequence per gene alignments; and finally, a *KIR* sequence imputation workflow was developed to enable alignment to any described *KIR* allele sequence. These innovations enable for the first time highly-automated, high-throughput *KIR* sequence analysis from short-read sequencing data, and importantly, largely obviate the need for user expertise in the *KIR* system.

## Materials and methods

The major innovations in PING, detailed below, include the use of multiple-sequence per gene alignment references that incorporate the allelic diversity of *KIR*, and genotype-matched alignment references (Fig 1). The use of a diverse reference set in the copy number module substantially improves copy number determination for *KIR2DL2*, *KIR2DL3*, *KIR2DS1* and *KIR2DL1* compared to our prototype approach. The improved performance enables an innovative alignment workflow that dynamically constructs genotype-matched alignment references based on the so-established gene content and a preliminary genotype determination. The genotypes determined by this novel genotype-aware alignment workflow are highly accurate, with few unresolved calls.

### Preprocessing the database

**Imputation of uncharacterized regions and extension of untranslated regions to generate comprehensive alignment reference sequence.** *KIR* allele sequences used throughout this workflow are provided by the Immuno Polymorphism Database—KIR (IPD-KIR), release 2.7.1 [37]. However, many *KIR* allele sequences in IPD-KIR have only been characterized for exons, indeed, 65% of named alleles have less than 20% of their full-length sequence

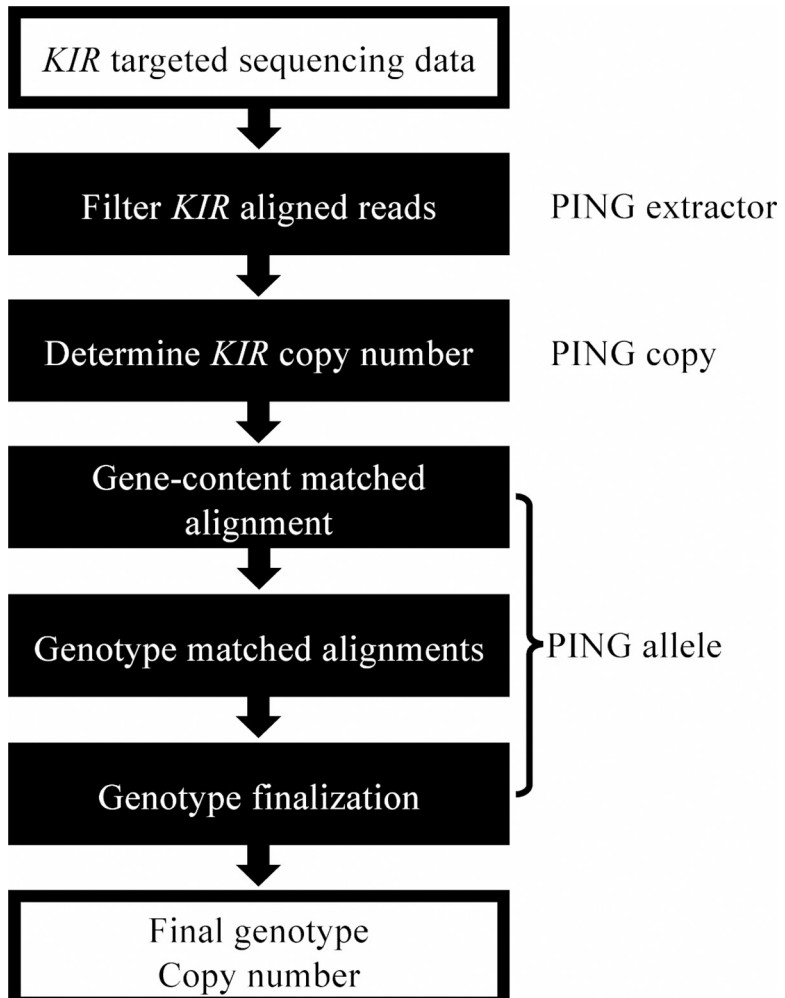

**Fig 1. Overview of the PING pipeline.** The PING pipeline processes *KIR* targeted sequencing data to determine *KIR* gene copy number and allele genotypes through a series of modules. First, *KIR* aligned reads are filtered through an alignment to a set of *KIR* haplotypes in PING extractor. Second, copy number of *KIR* genes are determined through an exhaustive alignment to a diverse set of *KIR* sequences in PING copy. Finally, PING allele performs a series of alignments to determine the most congruent *KIR* genotype, which informs a final round of alignment and genotype determination. Additionally, PING reports any identified novel SNPs and new alleles (SNP combinations not found in any described *KIR* allele sequence).

characterized (Fig 2A). Additionally, IPD-KIR allele sequences only include ~250bp of 5' untranslated region (UTR) sequence and ~500bp of 3' UTR sequence, reducing alignment depths across the first exon and potential regulatory regions. Thus, to maximize the utility of reference *KIR* sequences, we designed and implemented a protocol to impute the sequence of the intronic regions, followed by a protocol to extend UTRs to 1000bp each.

As reference sequences, we used all *KIR* alleles described in the IPD—KIR [21], release 2.7.1. A subset of these sequences is not completely characterized through all exons and introns. We therefore used gene-specific alignments of known sequences, provided by IPD-KIR as multiple sequence format (MSF) files, and completed each allele sequence to comprise the invariant nucleotides together with each variable position represented by an 'N'. Using this imputation method, we generated a new set of reference alleles in which ~90% of the 905 alleles were >98% complete (Fig 2B).

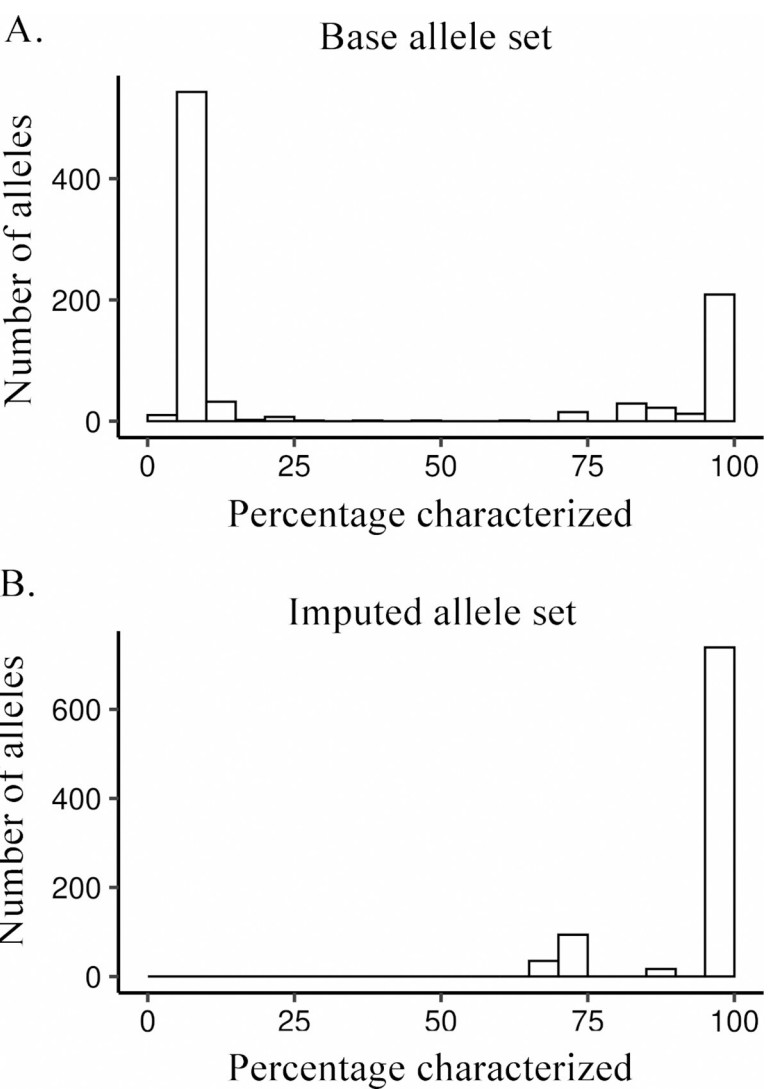

**Fig 2. *KIR* sequence characterization before and after imputation.** (A) Histogram of IPD-KIR allele sequence lengths, shown as percentage of longest sequence for each gene and major allele group. (B) Histogram of IPD-KIR allele sequence lengths after imputation, shown as percentage of longest sequence for each gene and major allelic group.

To extend UTR sequence, donor sequences, sourced from the full *KIR* haplotype sequences used in PING extractor, were appended to the ends of each reference sequence to generate 1000bp long UTRs. A single 3'UTR and 5'UTR donor sequence was used for each gene and major allelic group.

**Designing a minimized reference allele set.**   While performing a comprehensive alignment to the full *KIR* allele set reduces misalignments caused by reference sequence bias, it demands substantial resource utilization, as well as a large alignment and processing time cost which can prove untenable for processing large datasets. For example, copy determination processing for 10 paired-end sequences using 36 threads took 4.95 hours with a maximum Binary Alignment Map (BAM) [38] file size of 338.2MB.

To address this issue, we constructed a minimized set of reference alleles to improve resource utilization and processing time while still reducing misalignments caused by

reference sequence bias. The minimized reference set consists of five alleles for each *KIR* gene and major allelic group (S1 Table). The use of five alleles per gene was empirically determined to be sufficient for reducing reference sequence bias, while still considerably reducing the computational burden of multiple-sequence per gene alignments. Designing this reference set was guided by selecting alleles which had fully-characterized or nearly fully-characterized sequence, the secondary criteria was maximizing SNP diversity between the reference alleles of each gene, and third was selecting reference alleles to sequester reads susceptible to off-gene mapping. For example, reference sequences to represent *KIR2DS1*002* as well as *KIR2DL1*004* were selected to sequester reads that perfectly align to both. Notable characteristics of the reference set are the separation of *KIR2DL2* from *KIR2DL3*, the separation of *KIR3DL1* from *KIR3DS1*, and the merging of *KIR2DL5A* and *KIR2DL5B*.

## PING workflow methods

**Copy number determination–PING copy.**  The high sequence similarity between *KIR* genes coupled with extensive structural variation and nucleotide diversity makes copy number determination a non-trivial task. Our copy determination method is largely identical to that described in Norman et al. [27], in which copy number is determined by comparing the number of reads that align uniquely to each *KIR* gene across a batch of samples using *KIR3DL3* as a normalizer. The improvement made by our method is the use of a comprehensive *KIR* reference composed of 905 distinct sequences from the imputed and extended allele set, instead of a single-sequence per gene reference. The use of a comprehensive reference provides a more accurate comparison of the number of reads that align uniquely to any *KIR* gene.

**KIR virtual probes–PING allele.**  To determine the presence of target alleles or allelic groups that are prone to misidentification due to read misalignments, we have developed a set of virtual, or text-based, probes. The probe set includes those described in Norman et al. [27], as well as additional, custom probes (S2 Table). Probes are designed to match sequence that is unique to the target allele or allelic group, and sequence uniqueness is determined by a grep search over the imputed and extended IPD-KIR sequence set. Application of the probe set is performed using grep over the sequencing data, counting the number of unique reads that contain sequence perfectly matching the probe. A probe hit is determined using a threshold of 10 matching reads.

**Genotype matched alignment workflow–PING allele.**  The overall alignment strategy of PING is to reduce reference sequence bias through using multiple-sequence per gene references, and the use of references that reflect the gene content makeup or genotype makeup of a sample (Fig 3). Additionally, PING utilizes multiple rounds of alignment and genotype determination with varied processing parameters to reduce bias introduced by assumptions made during the processing workflow. The intermediate and final alignment and genotyping rounds are referred to as 'initial' and 'final', respectively, when differentiating processing parameters.

The first step is an alignment to a multiple-sequence per gene, gene content matched reference. This gene-content aware alignment workflow constructs individualized alignment references based on the presence of certain *KIR* genes: *KIR3DP1*, *KIR2DS2*, *KIR2DL23*, *KIR2DL5A*, *KIR2DL5B*, *KIR2DS3*, *KIR2DS5*, *KIR2DP1*, *KIR2DL1*, *KIR2DL5*, *KIR3DL1S1*, *KIR2DS4*, *KIR3DL2*, *KIR2DS1* and *KIR3DL3* (assumed always present [39]). Reference sequences are selected from the diverse, minimized reference sequence set described above. We have included an option to align to the full comprehensive allele set, but this is not default behavior as these alignments are time and resource intensive.

An exhaustive alignment, an alignment in which all qualified read mappings are recorded, is performed and aligned reads are processed and formatted according to the alignment processing workflow, detailed in S1 Text, selecting for reads that uniquely map to a gene or major

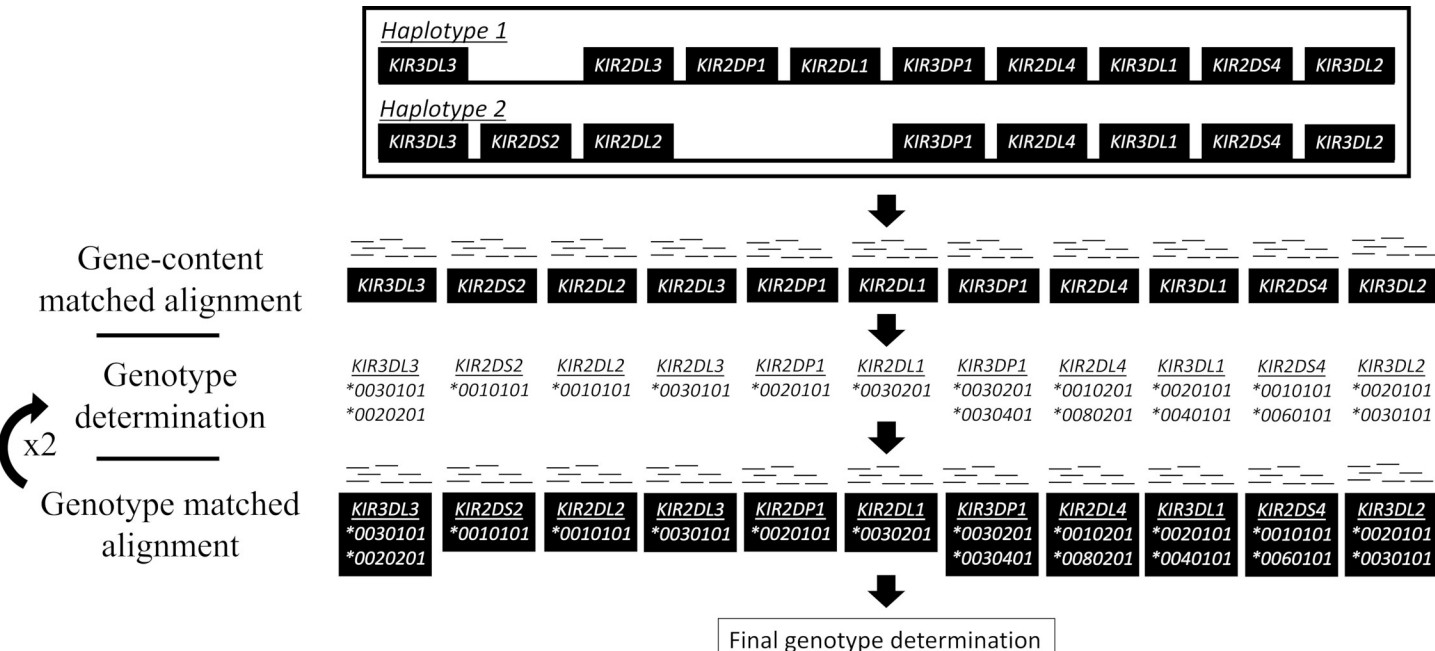

**Fig 3. Overview of the genotype aware alignment workflow.** Sequence gene content, determined by PING copy, informs the selection of reference sequence from a predefined set of diverse allele sequences. An exhaustive alignment is performed to the selected allele set, from which an initial genotype determination is made. The determined genotype informs selection of reference alleles for a genotype aware alignment, followed by another round of genotype determination. The genotype aware alignment and subsequent genotype determination is repeated, and the most congruent genotypings across all alignment rounds inform reference selection for a final round of alignment. A non-exhaustive alignment is performed to the selected allele set, from which all aligned reads are processed and used for the final genotype determination.

allelic group. The formatted uniquely-mapped read set is processed according to the genotype determination workflow, detailed below, to obtain an initial full-resolution genotype.

Second is a series of two alignments to genotype-matched references with varied processing parameters to identify the most congruent *KIR* genotype. Genotype congruence is determined by the least number of SNP mismatches between the determined allele typing(s) of a gene, and the aligned SNPs. For each genotype-matched alignment in this series, the determined allele typing(s), including any ambiguity, are used as reference sequence for the following alignment. For each alignment, genotypes are first determined at seven-digit (non-coding mutation level), then five-digit resolution (synonymous mutation level). This approach reduces the impact of uncharacterized regions of IPD-KIR allele sequences on genotype determination, as most sequences are fully characterized across exons. Genotype determination can be biased towards or against IPD-KIR alleles with uncharacterized regions depending on whether uncharacterized SNPs count as mismatches or not. To reduce time spent on genotype determination, any unambiguous typing that is perfectly matched to the aligned SNPs is locked in across all subsequent intermediate rounds of genotype determination.

The reference for the final alignment is built from the locked genotypings and the closest matched genotypings for genes without a locked genotyping. A non-exhaustive alignment is performed to the built reference, from which all aligned reads are processed and formatted according to the alignment processing workflow. The formatted read alignments are passed to the genotype determination workflow to obtain a final exonic (five-digit) resolution genotype.

**Additional methods utilized by the genotype matched alignment workflow–PING allele.** Genotype matched alignments and subsequent genotype determinations can get stuck on a mistyped allele due to persistent reference sequence bias. In other words, a false SNP call originating from misaligned reads can perpetuate itself in the genotype matched alignments

due to the same allele determination being made and the same alignment reference being used. To address this issue, we have included a method in the genotype matched alignments that will add the five allele sequences from the diverse, minimized reference set to the genotype-matched reference for any gene with an allele typing that does not perfectly match the aligned SNPs. The rationale behind this method is that mismatched allele typings are likely due to misaligned reads, and the use of the mismatched allele sequence as a reference will cause the read misalignments to be repeated in subsequent alignment and genotyping rounds. The addition of a diverse set of alignment sequences gives an avenue to break from this cycle by increasing the likelihood that a different allele typing will be made.

In building genotype-matched references PING allows any allele to be used as reference sequence, however, some allele sequences are only partially characterized even after imputation. The use of allele sequences containing uncharacterized sequence as alignment references can introduce reference sequence bias and drive read misalignments even if the reference alleles perfectly match the true genotype of the sample. To address this issue, we have included a method to add fully-characterized sequence to the alignment reference for any gene represented by only partially-characterized sequence(s). Fully-characterized alleles are pulled from the diverse, minimized reference set.

In the genotype-aware alignment workflow we found issues with false negative identifications of *KIR2DL1*004/*007/*010* due to reads cross-mapping to other gene sequences. This issue was rectified using virtual sequence probes specific to each of these *KIR2DL1* allele groups to identify *004/*007/*010* allele presence. If *KIR2DL1*010* is present, then the *KIR2DL1*010* allele sequence is added to the alignment reference. If *KIR2DL1*004* is present, then the *KIR2DL1*0040101* allele sequence is added to the alignment reference. If *KIR2DL1*007* is present, then the *KIR2DL1*007* allele sequence is added to the alignment reference. If multiple of these allele groups are present, then the *KIR2DL1*0040101* allele sequence is added to the alignment reference.

We implemented additional probes to identify alleles and structural variants prone to misidentification across *KIR2DL1*, *KIR2DL2*, *KIR2DL4*, *KIR2DS1*, *KIR3DP1* and *KIR3DS1*. For example, we implemented a probe to identify the *KIR2DL4* poly-A stretch at the end of exon 7, as well as a probe to identify *KIR3DP1* exon 2 deletion variants. The full list of probes used for reference refinement can be found in S2 Table.

**Genotype determination workflow–PING allele.** Indexed reads, detailed in S1 Text, are processed to generate a depth table spanning -1000bp 5'UTR to 1000bp 3'UTR for each *KIR* gene and major allelic group. Depths are marked independently for A, T, C, G, deletions and insertions. Depth tables are processed to generate SNP tables for positions passing a minimum depth threshold (default 8 for initial genotyping and 20 for final genotyping). To identify heterozygous positions the depth of each aligned variant is divided by the highest depth variant for that position, and up to three variants (A, T, C, G, deletions and insertions) passing the ratio threshold (default 0.25 for initial and final genotyping) are recorded.

Genotypes for each gene and major allelic group are determined from the aligned SNPs using a mismatch scoring approach. First, aligned homozygous SNPs are compared to each IPD-KIR allele, with SNP mismatches counting as a score of 1 and matches as 0. The lowest scoring alleles and alleles within a set scoring buffer of the lowest score (default of 4 for the initial genotyping workflow and 1 for the final genotyping workflow), are carried over into heterozygous position scoring.

For aligned heterozygous position scoring, all possible allele combinations are enumerated according to the determined copy of the gene under consideration, up to copy 3. For each aligned position, the variant(s) for each allele combination are compared to the aligned variants, with full matches counted as a score of 0 and mismatches scored according to the number

of mismatched variants. For each allele combination, the homozygous score of each component allele is added to the heterozygous score, and the lowest scoring combinations are returned as the determined genotype. For the final genotyping workflow, only perfectly scoring combinations are accepted, with any mismatches resulting in an unresolved genotype.

The same workflow is applied to both initial and final genotyping with some important distinctions. In the initial genotyping workflow, the imputed and extended IPD-KIR allele sequences are used for SNP comparisons, uncharacterized variants within the comparison sequences are marked as full mismatches, and all aligned allele-differentiating SNP positions passing the depth threshold are compared and used for scoring. In the final genotyping workflow, the unimputed IPD-KIR allele sequences are used for SNP comparisons, uncharacterized variants within the comparison sequences are marked as matches, and only aligned exonic SNP positions passing the depth threshold are compared and used for scoring.

The final exonic resolution genotypes are processed to add null alleles to the genotype string for genes with copy 0 or 1 and combine component allele typings for the major allelic groups *KIR2DL2* and *KIR2DL3*, *KIR3DL1* and *KIR3DS1*, and *KIR2DS3* and *KIR2DS5*.

### Workflow validation

***KIR* synthetic sequence dataset.** A *KIR* synthetic dataset consisting of 50 sequences was generated using the ART next-generation sequencing read simulator [40]. ART parameters were set to simulate 150-bp paired-end reads at 50x coverage, with a median DNA fragment length of 200 using quality score profiles from the HiSeq 2500 system. Eleven of the *KIR* haplotypes described in Jiang et al. [41] were used to simulate structural variation of the *KIR* region. Two of the eleven haplotypes were randomly selected with replacement to establish the copy number for each sample. Allele sequences were selected randomly without replacement from the imputed and extended set according to the copy number of each gene. Any uncharacterized regions in the selected allele sequences were replaced with sequence from a random fully-characterized sequence from the same gene. Reads were named according to the source allele, enabling tracing of misaligned reads to their source allele and gene. The full synthetic dataset is available at: https://github.com/wesleymarin/KIR_synthetic_data.

Discordant genotype results for the synthetic dataset were investigated by identifying the source gene for each read aligned to the incorrectly genotyped gene. The results were summarized to show the total number of reads from each source gene to each aligned gene, Table A in S3 Table, and a read sharing diagram was generated using the circlize [42] package in R [43].

**Characterization of *KIR* reference cohorts for PING development.** A significant barrier to the development of bioinformatic methods for high-resolution *KIR* sequence interpretation is the lack of a well-characterized reference cohort. Without such a resource it is extremely difficult to recognize and resolve issues with read misalignments, which can result in SNP calls that appear reasonable in many cases. To resolve this issue, we have characterized a *KIR* reference cohort of 379 healthy individuals of European ancestry that had been previously sequenced using our *KIR* target capture method [44], with the results meticulously curated by manual alignment and inspection of all sequences to provide a ground truth dataset to aid pipeline development (Table A in S4 Table). Furthermore, the European samples were independently sequenced and genotyped for *KIR* by our collaborators at the DKMS registry for volunteer bone marrow donors [31]. Any discordant typing or gene content results were resolved through direct examination of sequence alignments, and where necessary, confirmatory sequencing.

In order to validate our method on a second, divergent population, we also examined a previously characterized cohort of African Khoesan individuals [45], for which *KIR* alignments and genotypes were manually inspected (Table B in S4 Table).

**Copy number and genotype concordance calculations.** For the European cohort, any genotype containing an unresolved *KIR3DL3* genotyping, a genotype for which the aligned SNPs do not perfectly match currently described alleles, in the truth dataset were excluded from copy number and genotype concordance comparisons. There were 16 full genotypes excluded by this criterion. Additionally, any individual gene with an unresolved genotype in the truth dataset were excluded from copy number and genotype concordance comparisons. For the synthetic dataset there were no simulated novel alleles, so the full dataset was used for copy number and genotype concordance comparisons. For the Khoesan dataset any individual gene with an unresolved genotype in the truth dataset were excluded from genotype concordance comparisons but were included for copy number comparisons.

Copy concordance was calculated by directly comparing the determined copy values to the validation copy values (S5 and S6 Tables). Genotype concordance was calculated on a per-gene basis by comparing each component allele of the determined typing to the truth genotype.

**Code availability.** The PING pipeline is available at https://github.com/wesleymarin/PING [46] with the following open source license: https://github.com/wesleymarin/PING/blob/master/LICENSE. Scripts and datasets used for data analysis are available at https://github.com/wesleymarin/ping_paper_scripts. PING was developed in the R programming language and tested on a Linux system. Additional requirements are: Samtools v1.7 or higher [38], Bcftools v1.7 or higher, and Bowtie2 v2.3.4.1 or higher [47]. The synthetic KIR sequence dataset is available at https://github.com/wesleymarin/KIR_synthetic_data.

## Results

### Extensive sequence identity among *KIR* genes is a major barrier for interpreting short-read sequencing data

Read misalignments due to sequence identity across the *KIR* region are a persistent challenge in *KIR* bioinformatics and often lead to spurious genotyping results. To quantify the extent of sequence identity and inform our investigation of SNPs suspected to be originating from misaligned reads, we performed a shared *k*-mer analysis using all 905 described *KIR* allele sequences in the Immuno Polymorphism Database (IPD)—KIR [21], release 2.7.1. Here, we transformed allele sequences into all distinct subsequences of sizes 50, 150, and 250 to compare sequence identity between genes. Shared k-mer proportions were calculated by dividing the number of shared k-mers by the total number of k-mers of that gene. K-mer sharing diagrams were generated using the circlize [42] package in R [43].

Our analysis showed that many genes share significant sequence identity at sequencing lengths commonly used in next generation sequencing (NGS) technology (Fig 4A). For example, *KIR2DL5A* shares 12,591 of its 15,359 distinct 150-mers (82%) with *KIR2DL5B* (Fig 4B and S8 Table), making it extremely difficult to distinguish short reads originating from these genes. Likewise, over 90% of the distinct 50-mers, and over 50% of distinct 250-mers of *KIR2DS1* are shared with other genes, the vast majority with *KIR2DL1*. This analysis allowed us to identify specific "hotspots" for read misalignments, informing post-alignment modifications (detailed previously) to minimize their impact.

### Development of a comprehensive *KIR* alignment reference enables accurate copy number determination of *KIR2DL1*, *KIR2DS1*, *KIR2DL2* and *KIR2DL3*

We compared single-sequence per gene vs. multiple-sequence per gene reference alignments using the synthetic dataset. The reads from this dataset are labeled according to their source gene, providing a straightforward approach to quantify off-target alignments. This comparison

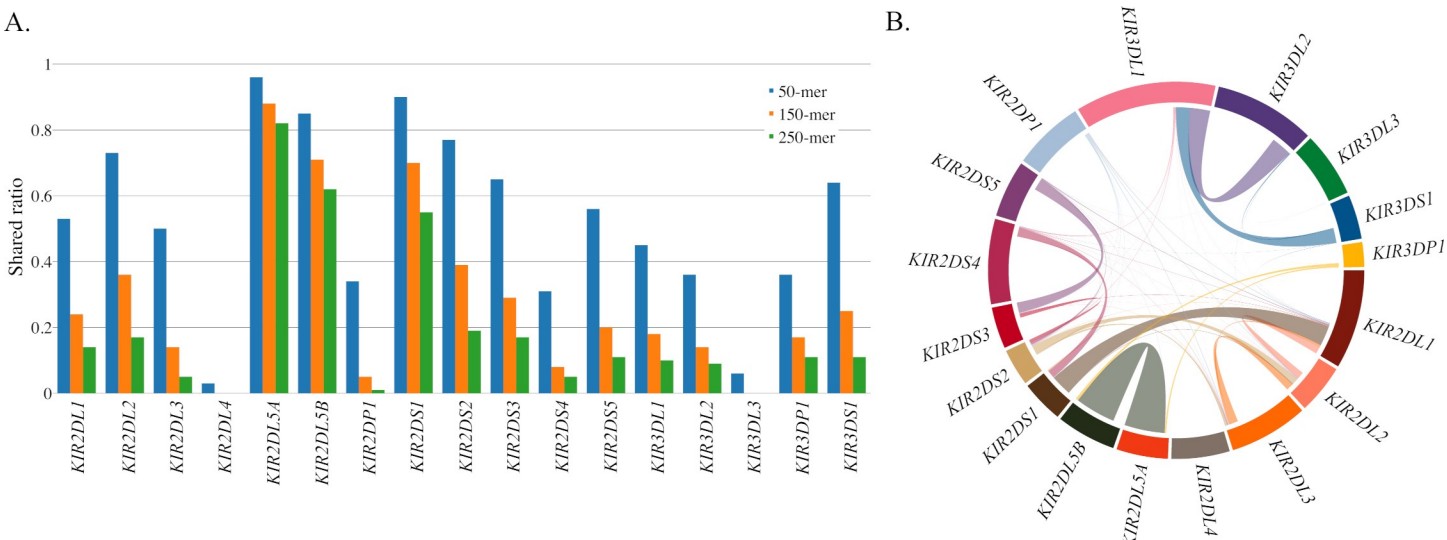

**Fig 4. K-mer analysis of *KIR* gene sequence similarity.** (A) Ratio of distinct k-mers of size 50, 150 and 250 that are shared between the indicated *KIR* gene and others. The inverse of these bars (not shown) would indicate the proportion of k-mers that are distinct to that gene and not found in the alleles of other genes. (B) 150-mer connections between *KIR* genes, the size of the connecting line roughly indicates the total number of shared 150-mers.

showed substantial reductions in the frequency of read misalignments (reads mapping to an off-target gene) across *KIR2DL1*, *KIR2DL23*, *KIR2DL5A/B*, *KIR2DS1*, *KIR2DS35*, and *KIR3DL1S1*, and small reductions for *KIR3DL2*, *KIR3DL3*, and *KIR3DP1* for the multiple-sequence per gene reference (Fig 5A and S9 Table). Applying the comprehensive reference allele set to gene content and copy number determination, we achieved significant improvement over a single-sequence per gene reference for *KIR2DL1*, *KIR2DS1* and the allelic groups *KIR2DL2* and *KIR2DL3* (S1–S3 Figs). The copy number for *KIR2DS1*, per example, which is highly prone to read misalignments due to similarity to *KIR2DL1* and *KIR2DS4* (Fig 4B), was clearly determined (Fig 5B).

## PING delivers accurate copy number and high-resolution allele calls

The overall performance of PING was assessed using our European *KIR* reference cohort, a synthetic *KIR* dataset, and a Khoesan *KIR* reference cohort. Results for PING copy number determination are summarized in Table 1, showing at least 97% concordance for the European cohort for all compared genes, with most genes exhibiting more than 99% concordance. Performance for the synthetic dataset showed 100% copy concordance for all compared genes except for *KIR2DL1*, at 98%, and *KIR2DS3*, at 92%. Finally, performance for the Khoesan cohort showed at least 95% concordance for all compared genes except for *KIR2DL2*, at 61%, *KIR2DL5A/B*, at 88%, *KIR2DS5*, at 89%, and *KIR2DL1*, at 94%. Across all datasets *KIR3DL3* was not compared due to its use as a reference gene, and for the European and Khoesan cohorts the pseudogene *KIR3DP1* was not compared due to an absence of validation data.

Performance of genotype determination was assessed at three-digit resolution (protein level) for the European and Khoesan cohorts, and at five-digit resolution (synonymous mutation level) for the synthetic dataset (Table 2). The results were categorized as genotype matches, mismatches or unresolved genotypes, which were cases where PING could not make a genotype determination.

PING genotype determination for the European cohort showed low percentages of unresolved genotypes with few mismatches for all compared genes except for *KIR2DP1*, with 10.3%

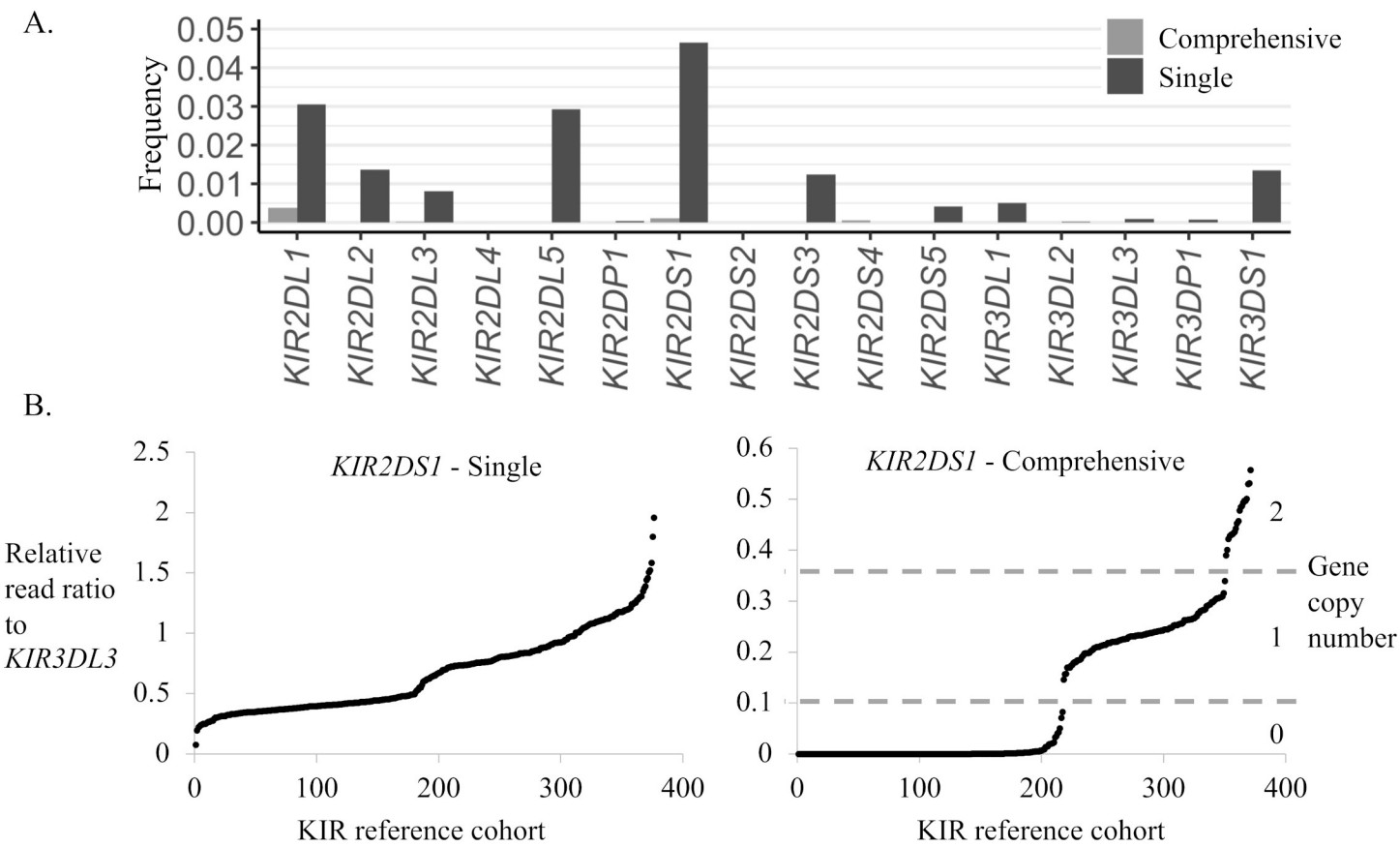

**Fig 5. Use of comprehensive reference improves copy determinations.** (A) Frequencies of off-target read mappings using a comprehensive reference vs. a single-sequence per gene reference for the synthetic dataset. (B) Single-sequence reference vs. comprehensive reference copy number plot of *KIR2DS1* for the European cohort. The copy plot of the single-sequence reference alignment shows no differentiation between copy groupings while the comprehensive reference alignment shows a clear distinction between the copy 0, 1 and 2 groups.

unresolved. Notable results for the European cohort were the low frequencies of unresolved genotypes across most genes, except *KIR2DP1*, and the extremely low frequencies of mismatched genotypes, below 1%, for 9 out of the 12 genes compared.

Determined genotypes for the synthetic dataset showed over 95% concordance for *KIR3DL3*, *KIR2DP1*, *KIR2DL4*, *KIR3DL1S1*, and *KIR3DL2*. However, the synthetic dataset showed high percentages of unresolved genotypes for *KIR2DL23*, *KIR3DP1* and *KIR2DS1*, each over 10% unresolved, and *KIR2DS35* and *KIR2DL1* showed 5.8% and 9.7% unresolved, respectively. *KIR2DL5A/B* and *KIR3DP1* showed the highest mismatched genotype percentages, at 10.6% and 5.5%, respectively, while *KIRDL3*, *KIR2DP1*, *KIR2DL4* and *KIR2DS1* each showed 0.0% mismatched genotypes.

Determined genotypes for the Khoesan cohort showed highly concordant genotypes for *KIR2DL4*, at 96.1%, and *KIR2DS1*, at 99.5%. Additionally, results for this dataset showed low mismatch frequencies for *KIR2DS2*, *KIR2DL23*, *KIR2DP1*, *KIR2DL1*, *KIR3DL1S1* and *KIR3DL2*, each below 5.0% mismatched. However, the Khoesan cohort showed moderate mismatch frequencies for *KIR3DL3*, at 6.2%, KIR2DL5A/B, at 7.1%, *KIR2DS35*, at 5.2%, and *KIR2DS4*, at 5.1%, and higher unresolved rates for *KIR2DS2*, *KIR2DL23*, *KIR2DP1*, *KIR2DL1*, *KIR2DS4* and *KIR3DL2*, each over 10.0% unresolved.

For the European and Khoesan cohorts the pseudogene *KIR3DP1* was not compared due to an absence of validation data.

**Table 1. Copy number determination performance.** Concordance table comparing copy numbers determined by PING for the European reference cohort, a synthetic *KIR* dataset, and a Khoesan reference cohort.

| Gene | European | N | Synthetic | N | Khoesan | N |
|:---:|:---:|:---:|:---:|:---:|:---:|:---:|
| *KIR3DL3* | - | - | - | - | - | - |
| *KIR2DS2* | 0.988 | 343 | 1.00 | 50 | 0.97 | 100 |
| *KIR2DL2* | 0.994 | 331 | 1.00 | 50 | 0.61 | 100 |
| *KIR2DL3* | 0.994 | 331 | 1.00 | 50 | 1.00 | 100 |
| *KIR2DL5A/B* | 0.997 | 343 | 1.00 | 50 | 0.88 | 100 |
| *KIR2DS3* | 0.988 | 343 | 0.92 | 50 | 0.97 | 100 |
| *KIR2DS5* | 0.985 | 342 | 1.00 | 50 | 0.89 | 100 |
| *KIR2DP1* | 0.982 | 338 | 1.00 | 50 | 0.95 | 100 |
| *KIR2DL1* | 0.970 | 334 | 0.98 | 50 | 0.94 | 100 |
| *KIR3DP1* | - | - | 1.00 | 50 | - | - |
| *KIR2DL4* | 0.994 | 341 | 1.00 | 50 | 1.00 | 100 |
| *KIR3DL1* | 0.997 | 340 | 1.00 | 50 | 1.00 | 100 |
| *KIR3DS1* | 0.997 | 339 | 1.00 | 50 | 0.99 | 100 |
| *KIR2DS1* | 0.988 | 342 | 1.00 | 50 | 1.00 | 100 |
| *KIR2DS4* | 0.991 | 343 | 1.00 | 50 | 0.99 | 100 |
| *KIR3DL2* | 0.988 | 326 | 1.00 | 50 | 1.00 | 100 |

Looking specifically at the concordance of resolved genotypes, the European cohort showed greater than 98.0% concordance across all compared genes except for *KIR2DL5A/B*, at 95.5%, and *KIR2DS4*, at 95.6% (Table 3). The synthetic dataset showed 100% concordance for *KIR3DL3*, *KIR2DP1*, *KIR2DL4*, *KIR3DL1S1* and *KIR2DS1*, over 95% concordance for *KIR2DS2*, *KIR2DL23*, *KIR2DS35*, *KIR2DL1*, *KIR2DS4* and *KIR3DL2*. The lowest performing genes in the synthetic dataset were *KIR2DL5A/B*, at 89%, and KIR3DP1, at 93%. The Khoesan cohort showed 100% concordance for *KIR2DS1*, over 95% concordance for *KIR2DS2*, *KIR2DL23*, *KIR2DP1*, *KIR2DL4*, *KIR3DL1S1* and *KIR3DL2*, and over 90% concordance for *KIR3DL3*, *KIR2DL5A/B*, *KIR2DS35*, *KIR2DL1* and *KIR2DS4*.

Together, these results demonstrate that PING accurately provides *KIR* genotyping across distinct populations.

## Analysis of discordant determined copy number and genotype results

The discordant copy results for *KIR2DS3* in the synthetic dataset were the result of poor differentiation between copy groups (S3 Fig). The highly discordant *KIR2DL2* copy number result for the Khoesan cohort was due to non-differentiable copy number groupings (S2 Fig). Since the *KIR2DL3* copy differentiation for this cohort was well defined, these results were used to set the *KIR2DL2* copy number prior to genotype determination using the formula KIR2DL2_copy = 2 – KIR2DL3_copy.

An investigation into the discordant genotypes for the synthetic dataset showed discordant genotype determination results for *KIR2DS35* were largely due to source reads from *KIR2DS3* aligning to *KIR2DS5* reference sequence, with a smaller number of reads from *KIR2DS5* aligning to *KIR2DS3* reference sequence (Fig 6 and Table A in S3 Table). This differential read flow between the two allelic groups is reflected in the component allele typings, with six discordant *KIR2DS5* genotypings and two discordant *KIR2DS3* genotypings (Table C in S7 Table). Intragenic misalignments are a product of how the PING workflow is structured, as major allelic groups, such as *KIR2DS3* and *KIR2DS5*, are treated as independent genes during alignment

**Table 2. Genotype determination performance.** Genotype determination performance table comparing the genotypes determined by PING to the validation genotypes for each dataset. Possible outcomes are 'Match', where the determined component allele matches the validation allele, 'Mismatch', where the determined component allele does not match the validation allele, or 'Unresolved', where PING was unable to determine a genotype, but the validation allele was not marked as unresolved. The coloring signifies concordance level, where green is 0–10% discordant, yellow is 10–15% discordant, and red is over 15% discordant.

| Gene | Dataset | Match | Mismatch | Unresolved | N |
|---|---|---|---|---|---|
| KIR3DL3 | European | 0.959 | 0.009 | 0.032 | 686 |
|  | Synthetic | 0.960 | 0.000 | 0.040 | 100 |
|  | Khoesan | 0.887 | 0.062 | 0.050 | 80 |
| KIR2DS2 | European | 0.975 | 0.012 | 0.013 | 686 |
|  | Synthetic | 0.940 | 0.040 | 0.020 | 100 |
|  | Khoesan | 0.835 | 0.005 | 0.160 | 188 |
| KIR2DL23 | European | 0.965 | 0.003 | 0.032 | 656 |
|  | Synthetic | 0.850 | 0.020 | 0.130 | 100 |
|  | Khoesan | 0.810 | 0.042 | 0.149 | 168 |
| KIR2DL5A/B | European | 0.927 | 0.044 | 0.029 | 687 |
|  | Synthetic | 0.856 | 0.106 | 0.038 | 104 |
|  | Khoesan | 0.857 | 0.071 | 0.071 | 126 |
| KIR2DS35 | European | 0.982 | 0.006 | 0.012 | 683 |
|  | Synthetic | 0.923 | 0.019 | 0.058 | 104 |
|  | Khoesan | 0.871 | 0.052 | 0.078 | 116 |
| KIR2DP1 | European | 0.890 | 0.007 | 0.103 | 672 |
|  | Synthetic | 0.971 | 0.000 | 0.029 | 103 |
|  | Khoesan | 0.721 | 0.012 | 0.267 | 86 |
| KIR2DL1 | European | 0.961 | 0.009 | 0.030 | 666 |
|  | Synthetic | 0.883 | 0.019 | 0.097 | 103 |
|  | Khoesan | 0.803 | 0.045 | 0.152 | 132 |
| KIR3DP1 | European | - | - | - | - |
|  | Synthetic | 0.773 | 0.055 | 0.173 | 110 |
|  | Khoesan | - | - | - | - |
| KIR2DL4 | European | 0.980 | 0.007 | 0.013 | 685 |
|  | Synthetic | 1.000 | 0.000 | 0.000 | 110 |
|  | Khoesan | 0.961 | 0.006 | 0.032 | 154 |
| KIR3DL1S1 | European | 0.962 | 0.006 | 0.032 | 686 |
|  | Synthetic | 0.955 | 0.000 | 0.045 | 110 |
|  | Khoesan | 0.873 | 0.028 | 0.099 | 142 |
| KIR2DS1 | European | 0.985 | 0.003 | 0.012 | 682 |
|  | Synthetic | 0.900 | 0.000 | 0.100 | 100 |
|  | Khoesan | 0.995 | 0.000 | 0.005 | 198 |
| KIR2DS4 | European | 0.943 | 0.044 | 0.013 | 685 |
|  | Synthetic | 0.940 | 0.020 | 0.040 | 100 |
|  | Khoesan | 0.793 | 0.051 | 0.157 | 198 |
| KIR3DL2 | European | 0.965 | 0.008 | 0.028 | 648 |
|  | Synthetic | 0.990 | 0.010 | 0.000 | 100 |
|  | Khoesan | 0.819 | 0.011 | 0.170 | 188 |

and genotyping. Intragenic misalignments were also a large contributor to *KIR3DL1S1* discordance, with reads supplied by *KIR3DL1* mapping to *KIR3DS1* reference sequence.

The analysis showed *KIR3DP1* as a major hub for receiving misaligned reads, with reads being contributed by each other *KIR* gene. In fact, *KIR3DP1* was largely the only receiver for

**Table 3. Resolved genotype concordance.** PING genotype determination performance for the European reference cohort, a synthetic *KIR* dataset, and the Khoesan reference cohort for each considered *KIR* gene.

| Gene | European | N | Synthetic | N | Khoesan | N |
|------|----------|---|-----------|---|---------|---|
| *KIR3DL3* | 0.991 | 664 | 1.00 | 96 | 0.934 | 76 |
| *KIR2DS2* | 0.988 | 677 | 0.96 | 98 | 0.994 | 158 |
| *KIR2DL23* | 0.997 | 635 | 0.98 | 87 | 0.951 | 143 |
| *KIR2DL5A/B* | 0.955 | 667 | 0.89 | 100 | 0.923 | 117 |
| *KIR2DS35* | 0.994 | 675 | 0.98 | 98 | 0.944 | 107 |
| *KIR2DP1* | 0.992 | 603 | 1.00 | 100 | 0.984 | 63 |
| *KIR2DL1* | 0.991 | 646 | 0.98 | 93 | 0.946 | 112 |
| *KIR3DP1* | - | -k | 0.93 | 91 | - | - |
| *KIR2DL4* | 0.993 | 676 | 1.00 | 110 | 0.993 | 149 |
| *KIR3DL1S1* | 0.994 | 664 | 1.00 | 105 | 0.969 | 128 |
| *KIR2DS1* | 0.997 | 674 | 1.00 | 90 | 1.000 | 197 |
| *KIR2DS4* | 0.956 | 676 | 0.98 | 96 | 0.940 | 167 |
| *KIR3DL2* | 0.992 | 630 | 0.99 | 100 | 0.987 | 156 |

misaligned reads originating from *KIR3DL3*, *KIR3DL2*, *KIR2DP1* and *KIR2DL4*. While the only genes receiving reads sourced from *KIR3DP1* were *KIR2DL5A* and *KIR2DL5B*.

The analysis also showed several gene pairings, where two genes largely sent and received reads from one another. Once such pairing was between *KIR2DL1* and *KIR2DS1*, where each gene were the largest contributor and receiver of reads for each other. Another pairing was between *KIR2DL2* and *KIR2DS2*, although both genes sent and received reads from several other genes.

This analysis illustrates the complex and highly interconnected nature of *KIR* and highlights the difficulty behind accurate interpretation of *KIR* short-read sequencing data.

## Performance

The run time and resource utilization of the PING pipeline was measured on an Intel Xeon 2.20 GHz CPU using 36 threads. For ten sequences from the synthetic dataset, it took 1.92 mins for *KIR* read extraction, 34.7 mins for copy determination aligning to the minimized reference set, and 2.10 hours for genotype determination aligning to the minimized reference set. The output directory size was 1.4GB.

## Discussion

Our shared k-mer analysis of all documented *KIR* variation shows the high degree of sequence identity between *KIR* genes and illustrates the challenges imposed by the homology of *KIR* on short-read interpretation workflows. It demonstrates that some genes are more likely to exhibit read misalignment problems than others. *KIR2DP1*, *KIR3DL3* and *KIR2DL4* have relatively unique sequence, while *KIR2DS1*, *KIR2DL5A* and *KIR2DL5B* have considerable shared sequence. This type of analysis provides an informative tool for investigating irregularities in the processing of *KIR* sequence data, revealing which genes are likely to be erroneously interpreted due to read misalignments for common sequencing read lengths. While paired-end sequencing with longer reads can improve read alignment fidelity, in our own experience 290bp paired-end reads with a median insert length of approximately 600bp still exhibited considerable read misalignment problems. It is important to note that this analysis does not account for unknown variation or intergenic sequence, two other sources of sequence variation that could potentially result in misaligned reads.

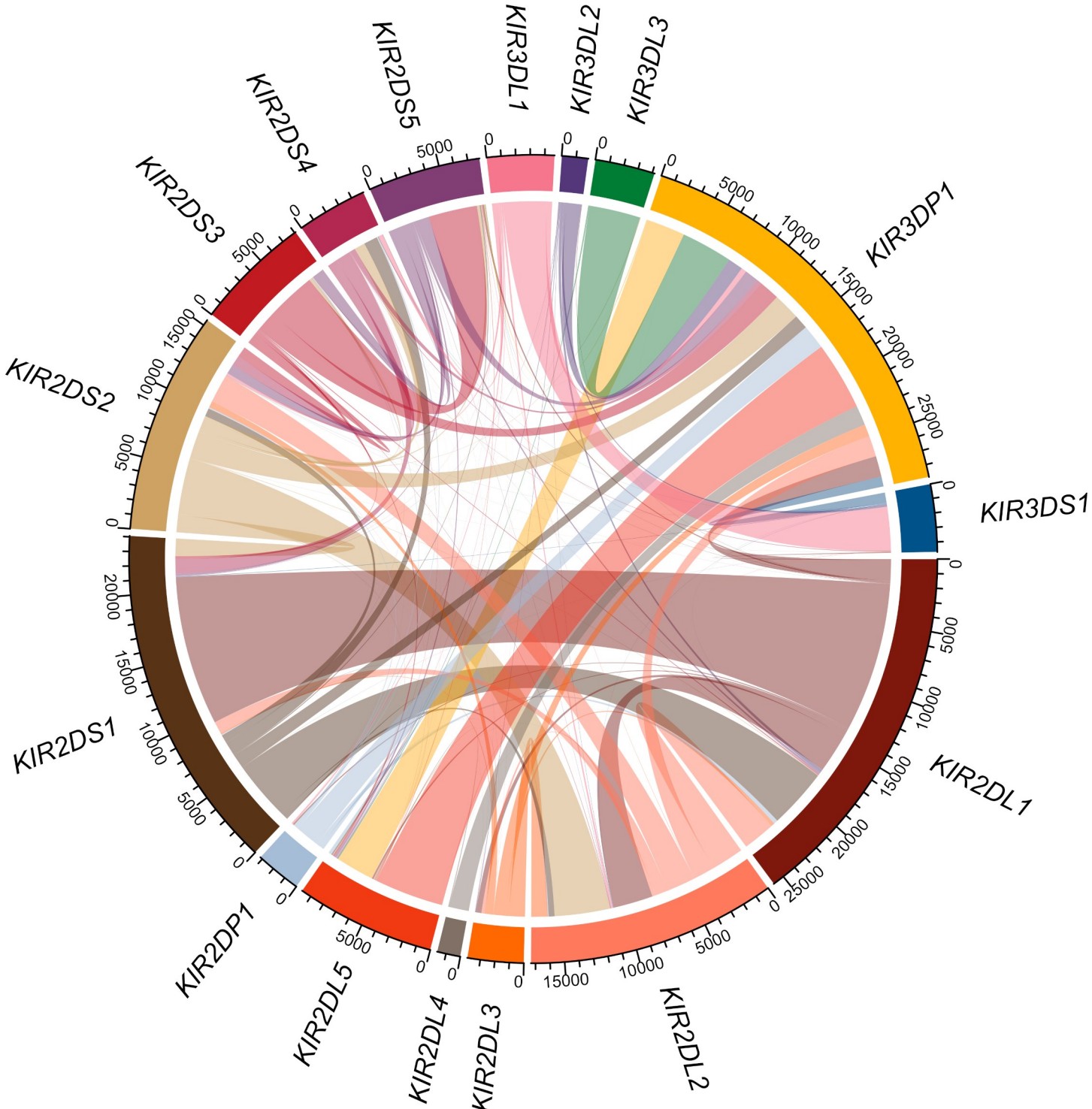

**Fig 6. Misaligned read sources in the synthetic dataset.** Analysis of mismatched or unresolved genotype determination results for the synthetic sequence dataset where all misaligned reads are traced back to their source gene. The connections between genes represent the number of misaligned reads, and the color of the connection represents the source gene.

An initial determination of *KIR* gene content and copy number provides an informative scaffold for minimizing misalignments through the exclusion of reference sequence representing absent genes, as well as a system for identifying misalignments by searching for erroneously-called heterozygous SNP alleles in hemizygous genes. Thus, accurate copy number determination is a vital first step in interpreting *KIR* sequencing data. To achieve this goal, we developed a copy number determination method in PING that uses all described *KIR* alleles as an alignment resource, increasing the total number of reference sequences from 15 to 905 compared to single-sequence per gene alignments. While many of these alleles were only defined across exonic regions, rendering them ineffective for short-read alignment, we developed and implemented a protocol for intronic region imputation. The imputation method cannot resolve all uncharacterized nucleotide sequence, yet it accounts for the majority of missing sequence, greatly increasing the number of useful reference alleles. The exhaustive alignment provides a comprehensive map of the alleles to which a read may align, facilitating copy number resolution of important *KIR* allelic groups and genes that share extensive sequence similarity, such as *KIR2DL2*, *KIR2DL3*, *KIR2DL1* and *KIR2DS1*, which were inaccessible to previous bioinformatic methods [27,48]. Additionally, the limited range of described UTR sequence, ~250bp 5'UTR and ~500bp 3'UTR, can reduce alignments over the first exon and potential regulatory regions [49,50].

The improved copy determination performance of PING, in addition to the expanded useful reference sequence repertoire, enables a smart, genotype-aware alignment workflow, designed to minimize read misalignments by closely matching reference sequences to the gene sequences present in the sequencing data. This alignment strategy addresses a major weakness of the filtration alignments utilized in the prototype workflow, which apply filters to retain gene-specific reads and eliminate cross-mapping reads regardless of the gene-content or sequence makeup, and thus often suffer from either inadequate or patchy aligned read depths after filtration. There is a valid concern about carrying forward alignment biases in the genotype-aware alignment workflow, and the *KIR* system in particular is sensitive to reference bias because the combination of highly polymorphic genes and high sequence similarity between genes means that small changes in the reference sequence can have large impacts on read alignments. We have implemented several methods to counteract potential alignment biases that could be carried forward by the genotype-aware alignments. The first is the use of virtual probes to identify alleles and structural variants prone to misidentification. The second is the addition of a curated sequence set to the alignment reference for any gene with a determined genotype that does not perfectly match the aligned SNPs, these sequences were selected to cover a large amount of the allelic diversity of the corresponding gene. Even with these countermeasures we still encounter some improper novel genotype determinations, likely due to reference sequence bias. Since no novel genotypes were simulated for the synthetic dataset the Synthetic data in the Unresolved column of Table 2 represent improper novel genotype determinations. Despite these limitations, the genotype-aware workflow achieves highly accurate genotype determinations for the European dataset (Table 2), and highly accurate resolved genotype determinations across all tested datasets (Table 3).

Both the synthetic dataset and Khoesan cohort showed higher levels of unresolved genotypes compared to the European cohort (Table 2). These datasets represent challenging data to correctly interpret, with the Khoesan being an extremely divergent population with many unresolved genotypes in the validation data, and the synthetic dataset consisting of random alleles, some of which used imputed sequence. An analysis into the discordant results for the synthetic dataset (Fig 6 and S3 Table) showed a complex web of cross-mapped reads. These cross-mapped reads can be extremely difficult to resolve because the high-degree of sequence shared among *KIR* genes (Fig 4) makes it almost impossible to determine correct mappings. Additionally, measures meant to prevent read misalignments, such as the use of virtual probes

to refine reference sequence selection, can serve as a double-edged sword, where the issue at hand is addressed but the changes create new sources for read misalignments.

An analysis into the discordant copy results highlights a major outstanding problem with the PING workflow since accurate copy determination is a central component of effective genotype-aware alignments, and the need for manual thresholding between copy groups introduces the component of user error. Continued development of the pipeline will address methods for automating copy determination for targeted sequencing data that matches or surpasses the accuracy achieved by manual thresholding. To compare PING against an existing method, we benchmarked against KPI [35] for determining *KIR* gene content (S10 Table), achieving 100% concordance for *KIR3DP1*, *KIR2DL3*, *KIR2DL4*, *KIR3DL3* and *KIR3DL2*, over 97% concordance for *KIR2DS5*, *KIR2DP1*, *KIR2DS3*, *KIR2DS2*, *KIR3DL1*, *KIR2DL2*, *KIR2DS4*, *KIR2DL1*, and *KIR2DL5A/B*, and over 95% concordance for *KIR3DS1*, and *KIR2DS1*.

We believe improved interpretation of *KIR* sequencing data will ultimately be achieved through longer-range sequencing technologies that can extend past the range of the shared sequence motifs, and through better imputation approaches that can more fully characterize currently described *KIR* alleles to provide a more robust alignment reference. While long-read data from a platform with cost-effective methods might be difficult to interpret due to the high error rates [51], the combination of long-reads and short-reads would cover the weaknesses of the respective technologies and should provide a highly accurate *KIR* interrogation method, indeed, long-read methods have provided valuable insights into *KIR* haplotypes [28]. We have not had the opportunity to test long-read technologies, but we anticipate a need for careful consideration of potential read misalignments when aligning the short and long reads together. Higher fidelity methods are currently under development, but currently the cost is prohibitive for the kind of high-throughput studies that PING was designed to address. Meanwhile, for samples for which genotypes are not easily resolvable, we recommend direct visualization of sequence alignments potentially coupled with alternative laboratory methods to more precisely determine genotypes.

While the PING workflow is specific to interpreting sequence originating from the *KIR* complex, the underlying strategies can be extended to other problematic genomic regions. For example, multiple-sequence per gene alignment strategies provide information for discriminating between reads derived from genes with high sequence identity and extensive nucleotide polymorphisms. Additionally, genotype-aware alignment strategies reduce bias introduced by the reference sequence for reads derived from genomic regions with high structural variation.

In conclusion, PING incorporates these innovations to provide accurate, high-throughput interpretation of the *KIR* region from short-read sequencing data. Together, these modifications provide a consistent *KIR* genotyping pipeline, creating a highly automated, robust workflow for interpreting *KIR* sequencing data. To the best of our knowledge, this is the only bioinformatic workflow currently available for high-resolution *KIR* genotyping from short-read data. Given the importance of KIR variation in human health and disease, availability of a highly accurate method to assess *KIR* genotypic variation should promote important discoveries related to this complex genomic region.

## Supporting information

**S1 Fig. European cohort copy determinations.**
(EPS)

**S2 Fig. Khoesan cohort copy determinations.**
(EPS)

**S3 Fig. Synthetic dataset copy determinations.**
(EPS)

**S1 Table. Diverse and minimized reference allele set.**
(XLSX)

**S2 Table. Virtual probe table for reference modifications.**
(XLSX)

**S3 Table. Discordant genotype analysis for synthetic dataset.**
(XLSX)

**S4 Table. Validation genotype table.**
(XLSX)

**S5 Table. PING determined copy number table.**
(XLSX)

**S6 Table. Validation copy number table.**
(XLSX)

**S7 Table. PING determined genotype table.**
(XLSX)

**S8 Table. K-mer gene match table.**
(XLSX)

**S9 Table. Synthetic dataset off-target read mappings.**
(XLSX)

**S10 Table. Benchmarking PING and KPI gene content performance.**
(XLSX)

**S1 Text. Genotype determination supporting methods.**
(DOC)

## Acknowledgments

We would like to thank Michael Wilson and Mark Seielstad for constructive comments.

## Author Contributions

**Conceptualization:** Wesley M. Marin, Paul J. Norman, Jill A. Hollenbach.

**Data curation:** Wesley M. Marin, Ravi Dandekar, Danillo G. Augusto, Tasneem Yusufali, Bianca Heyn, Jan Hofmann, Vinzenz Lange, Jürgen Sauter, Paul J. Norman, Jill A. Hollenbach.

**Formal analysis:** Wesley M. Marin.

**Funding acquisition:** Jill A. Hollenbach.

**Investigation:** Wesley M. Marin, Paul J. Norman.

**Methodology:** Wesley M. Marin, Paul J. Norman.

**Project administration:** Jill A. Hollenbach.

**Resources:** Paul J. Norman, Jill A. Hollenbach.

**Software:** Wesley M. Marin, Ravi Dandekar, Tasneem Yusufali.

**Supervision:** Jill A. Hollenbach.

**Validation:** Paul J. Norman.

**Visualization:** Wesley M. Marin.

**Writing – original draft:** Wesley M. Marin.

**Writing – review & editing:** Wesley M. Marin, Danillo G. Augusto, Paul J. Norman, Jill A. Hollenbach.

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
