## [Decision Letter · Decision Letter 0]

5 May 2021

Dear Dr. Hollenbach,

Thank you very much for submitting your manuscript "High-throughput Interpretation of Killer-cell Immunoglobulin-like Receptor Short-read Sequencing Data with PING" for consideration at PLOS Computational Biology.

As with all papers reviewed by the journal, your manuscript was reviewed by members of the editorial board and by several independent reviewers. In light of the reviews (below this email), we would like to invite the resubmission of a significantly-revised version that takes into account the reviewers' comments. Specifically, we would like the issue of comparison to two recently published tools brought by both reviewers to be fully addressed as well as other major concerns raised. 

We cannot make any decision about publication until we have seen the revised manuscript and your response to the reviewers' comments. Your revised manuscript is also likely to be sent to reviewers for further evaluation.

Sincerely,

Ferhat Ay, Ph.D

Associate Editor

PLOS Computational Biology

Jian Ma

Deputy Editor

PLOS Computational Biology

Reviewer's Responses to Questions

**Comments to the Authors:**

Reviewer #1: Authors describe a new version of the PING tool for genotyping of killer cell immunoglobulin-like receptor (KIR) genes that are critically important in immunogenomics studies and applications. Authors benchmark PING on many reference datasets and demonstrate its accuracy. While I am positive that this manuscript will present an interest for readers of PLOS Computational Biology, I highly recommend addressing the following comments regarding the description of the pipeline and benchmarking against the existing tools prior to publication:

1. I find the placement of the “K-mer similarity analysis” section confusing. Seems like the purpose of this section is to introduce the definition of shared k-mers that is used for demonstration of high repetitiveness of KIR genes, and the k-mer analysis itself is not a part of the PING tool. If so, I recommend combining this section with the section in Results as the definition of shared k-mers is pretty straightforward and probably shouldn’t take the whole section of the Methods.

2. It is difficult to create the correspondence between pipeline blocks in Figure 1 and sections of the Methods. Could you please add names of blocks from Figure 1 to the titles of sections to make understanding the pipeline easier?

3. Sections describing preprocessing of the KIR gene database and the pipeline are currently intermixed - e.g., the “Copy number determination workflow” section corresponding to the pipeline is located between sections describing manipulations with the database. Since it complicates following the Methods, I recommend making a separate section “Preprocessing the database” and make sections “Imputation of uncharacterized regions and extension of untranslated regions to generate comprehensive alignment reference sequence” and “Designing a minimized reference allele set” its subsections.

4. Line 223: initial genotyping and final genotyping are not introduced before and their purposes are unclear. Are they explained later in lines 274-289? In any case, add a short description of these steps before using them.

5. The overall description of the PING pipeline largely overlaps with the description provided in Norman et al., AJHG, 2016. Please provide a more specific description of the differences from the previous work in addition to one provided in lines 88-98.

6. Could you please comment on using long read technologies for genotyping and haplotyping KIR loci? Considering their relatively small sizes, I can imagine that a 70 kbp locus can be completely spanned by a single Pacbio / Nanopore read. Have you tried to use long read data in addition to short reads?

7. Have you benchmarked the new version of PING against recently released tools by Jieming Chen et al., Briefings in Bioinformatics, 2020 (doi.org/10.1093/bib/bbaa223) and Roe and Kuang, Front Immunol, 2020 (doi.org/10.3389/fimmu.2020.583013)? Please also add citations of these papers to the Introduction.

Reviewer #2: The manuscript describes a new tool for determination genotype and copy number for KIR genes. The tool is of huge interest and beautifully tackled the reference alignment part. It is also impressive to see that the tool is further applied to real-life scenario to see KIR genes copy number in population cohorts.

Strengths:

The manuscript stick to the theme in defining the problem and presenting a viable solution to it.

The underlying codes and reference sequences are made available through github repo.

Comments:

1. The workflow is interesting, however there seems to be no source of ground truth. Since, the authors have validated it in a simulated data, how we can be sure about the accuracy values?

2. While determining the genotypes, the authors are detecting the individual alleles and there are a few tools available for determine alleles in KIR data, the authors have not presented any comparison or references to those tools.

3. It is shown in the Figure 4 that even with 250-mer, few of the genes shared a high percentage with related genes. In such a case where the short read size is around 250 bases, how come any reliable selection of gene could be made?

4. One of the claim in the manuscript is that the reference alignment is improved in the workflow because of using an allele diversity, instead of single gene reference. However, no data is shown for improvement, and whether it is significant.

5. Genotype aware alignment might be carrying forward some biases and there would be issues in working with novel genotypes. Authors have not commented anything on this point.

**Have the authors made all data and (if applicable) computational code underlying the findings in their manuscript fully available?**

Reviewer #1: Yes

Reviewer #2: Yes

PLOS authors have the option to publish the peer review history of their article (what does this mean?). If published, this will include your full peer review and any attached files.

Reviewer #1: No

Reviewer #2: No
---

## [Decision Letter · Decision Letter 1]

16 Jul 2021

Dear Dr. Hollenbach,

We are pleased to inform you that your manuscript 'High-throughput Interpretation of Killer-cell Immunoglobulin-like Receptor Short-read Sequencing Data with PING' has been provisionally accepted for publication in PLOS Computational Biology.

Best regards,

Ferhat Ay, Ph.D

Associate Editor

PLOS Computational Biology

Jian Ma

Deputy Editor

PLOS Computational Biology

Reviewer's Responses to Questions

**Comments to the Authors:**

Reviewer #2: The authors have positively address the comments raised to them. They have also revised the manuscript to adapt those suggestions/comments. Furthermore, I am glad to see that the authors have acknowledged in the discussion section about possible biases or any limitations of the study or the field because of lack of ground truth data. I am pleased to recommend the acceptance of this revised article.

**Have the authors made all data and (if applicable) computational code underlying the findings in their manuscript fully available?**

Reviewer #2: Yes

PLOS authors have the option to publish the peer review history of their article (what does this mean?). If published, this will include your full peer review and any attached files.

Reviewer #2: No

---

## [Editor Report · Acceptance letter]

28 Jul 2021

PCOMPBIOL-D-21-00506R1 

High-throughput Interpretation of Killer-cell Immunoglobulin-like Receptor Short-read Sequencing Data with PING

Dear Dr Hollenbach,

I am pleased to inform you that your manuscript has been formally accepted for publication in PLOS Computational Biology. Your manuscript is now with our production department and you will be notified of the publication date in due course.

With kind regards,

Zsofi Zombor
